# Cryo-EM structure of catalytic ribonucleoprotein complex RNase MRP

Anna Perederina[1], Di Li[1], Hyunwook Lee[1], Carol Bator[1], Igor Berezin[1], Susan L. Hafenstein[1,2] & Andrey S. Krasilnikov [1,3✉]

RNase MRP is an essential eukaryotic ribonucleoprotein complex involved in the maturation of rRNA and the regulation of the cell cycle. RNase MRP is related to the ribozyme-based RNase P, but it has evolved to have distinct cellular roles. We report a cryo-EM structure of the *S. cerevisiae* RNase MRP holoenzyme solved to 3.0 Å. We describe the structure of this 450 kDa complex, interactions between its components, and the organization of its catalytic RNA. We show that some of the RNase MRP proteins shared with RNase P undergo an unexpected RNA-driven remodeling that allows them to bind to divergent RNAs. Further, we reveal how this RNA-driven protein remodeling, acting together with the introduction of new auxiliary elements, results in the functional diversification of RNase MRP and its progenitor, RNase P, and demonstrate structural underpinnings of the acquisition of new functions by catalytic RNPs.

[1] Department of Biochemistry and Molecular Biology, Pennsylvania State University, University Park, 16802 PA, USA. [2] Department of Medicine, Pennsylvania State University, Hershey, 17033 PA, USA. [3] Center for RNA Biology, Pennsylvania State University, University Park, 16802 PA, USA. ✉email: ask11@psu.edu

Ribonuclease (RNase) MRP, a site-specific endoribonuclease, is a ribonucleoprotein complex (RNP) comprising a catalytic RNA moiety and multiple (ten in *Saccharomyces cerevisiae*) protein components[1–4]. RNase MRP is an essential eukaryotic enzyme that has been found in practically all eukaryotes analyzed[5]. RNase MRP is localized to the nucleolus and, transiently, to the cytoplasm[3]. Known RNase MRP functions include its participation in the maturation of rRNA and in the metabolism of specific mRNAs involved in the regulation of the cell cycle[6–13]. Defects in RNase MRP result in a range of pleiotropic developmental disorders in humans[14].

RNase MRP is evolutionarily related to RNase P, a ribozyme-based RNP primarily involved in the maturation of tRNA[15–17]. RNase MRP appears to have split from the RNase P lineage early in the evolution of eukaryotes, acquiring distinct substrate specificity and cellular functions[5,18,19].

The catalytic (C-) domain of RNase MRP RNA (Fig. 1a) has the secondary structure resembling that of the C-domain of RNase P (Fig. 1b) and includes elements forming a highly conserved catalytic core[3–5]. The specificity (S-) domain of RNase MRP RNA does not have any apparent similarities with the S-domain of RNase P (Fig. 1a, b). Crosslinking studies[20] indicate the involvement of the RNase MRP S-domain in substrate recognition.

Most of the RNase MRP protein components are also found in eukaryotic RNase P[2]; the structures of *S. cerevisiae* and human

RNases P have been determined[21,22]. *S. cerevisiae* RNase MRP and RNase P share eight proteins (Pop1, Pop3, Pop4, Pop5, Pop6, Pop7, Pop8, and Rpp1 (two copies)); RNase MRP protein Snm1[23] has a homolog in RNase P (Rpr2), while Rmp1[24] is found only in RNase MRP. Shared proteins bind to both C- and S-domains[21,25,26]. Yeast RNase MRP proteins Pop1, Pop3, Pop4, Pop5, Pop6, Pop7, Pop8 have homologs in human RNase P[3,4,18,22], while Pop3, Pop4, Pop5, and Rpp1 have homologs in archaeal RNases P[3,4,27].

RNase MRP proteins Pop1, Pop6, Pop7 are also an essential part of yeast telomerase, where they are involved in the localization of the enzyme and form a structural module that stabilizes the binding of telomerase components Est1 and Est2[28,29].

Here, we report a cryo-EM structure of the *S. cerevisiae* RNase MRP holoenzyme solved to a nominal resolution of 3.0 Å. We reveal the overall architecture of the RNP, the structural organization of its catalytic RNA moiety, the substrate binding pocket of the enzyme, and interactions between RNase MRP components. Further, we compare the structure of RNase MRP to the structure of the progenitor RNP, eukaryotic RNase P. We show that, surprisingly, several of the proteins shared by RNase MRP and RNase P undergo RNA-driven structural remodeling, allowing the same proteins to function in distinct structural contexts. We demonstrate that while the structure of the catalytic center of RNase MRP is practically identical to that of RNase P, the topology of the substrate binding pocket of RNase MRP diverges

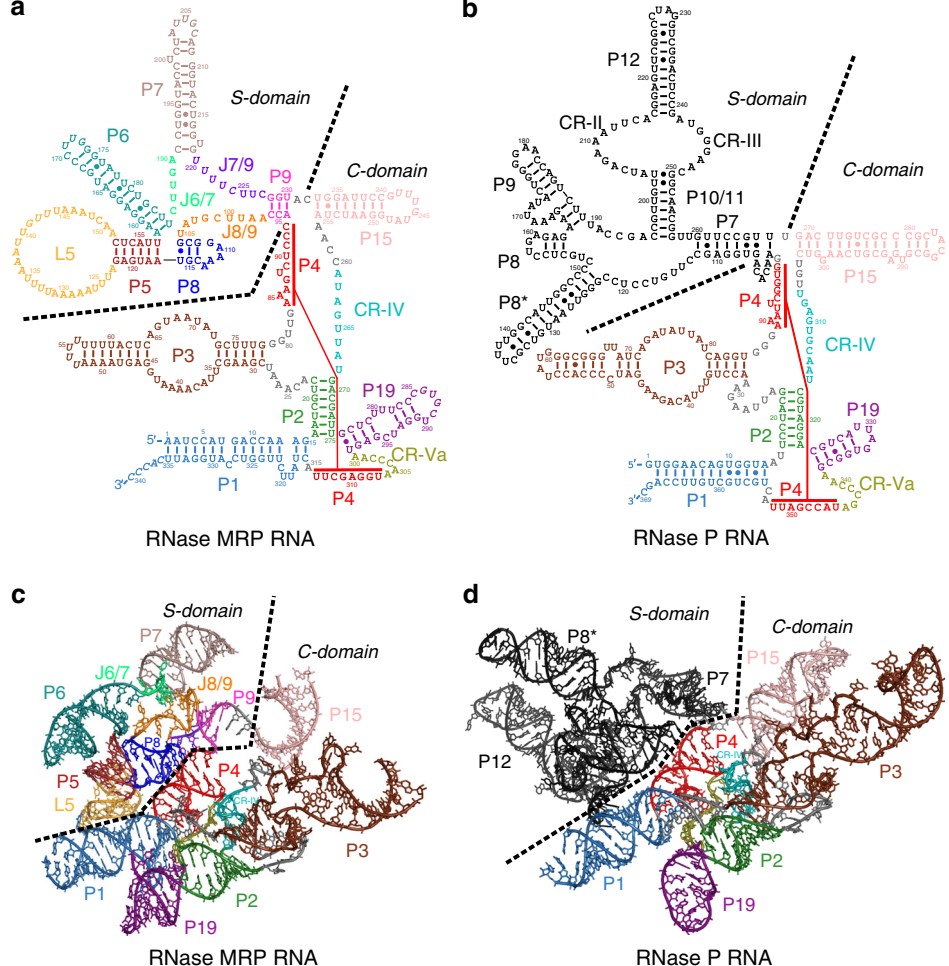

**Fig. 1 RNA components of RNase MRP and RNase P.** The catalytic (C-) domains of the two related enzymes are similar both in their secondary structures and in their folds, whereas the specificity (S-) domains are distinct. **a, b** Secondary structure diagrams of the RNase MRP and RNase P RNAs, respectively. **c, d** Folding of the RNase MRP and RNase P[21] RNAs, respectively, color coded as in (**a, b**).

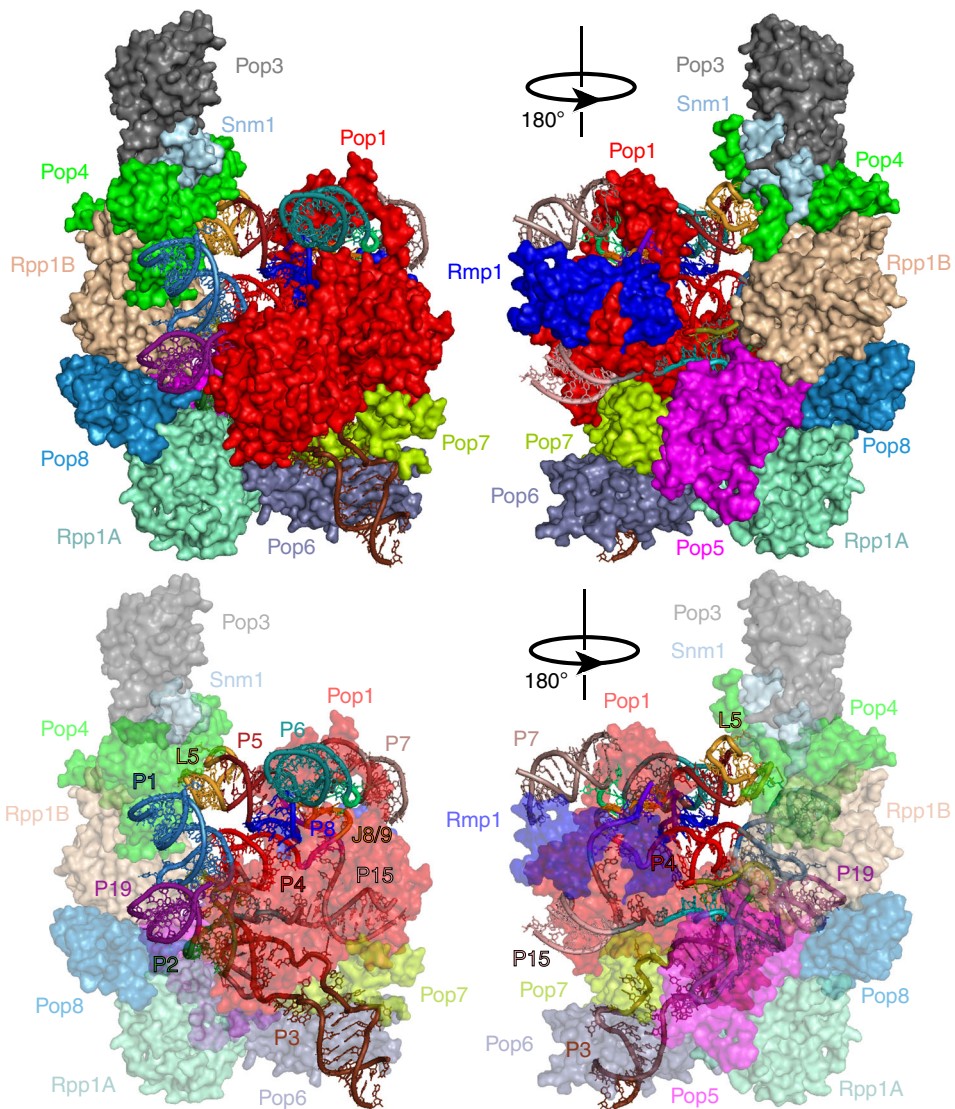

**Fig. 2 Structure of the RNase MRP holoenzyme.** Protein components (shown as surfaces) are color coded as marked; the RNA elements (shown as a cartoon) are color coded according to Fig. 1a.

from that of RNase P due to the presence of auxiliary RNA elements positioned in the immediate vicinity of the conserved catalytic center, due to the binding of RNase MRP protein Rmp1 near the catalytic center, as well as due to RNA-driven protein remodeling.

## Results and discussion

**Overall structure of RNase MRP.** RNase MRP holoenzyme used in the final 3D reconstruction was isolated from *S. cerevisiae* as a 1:1 mix with RNase P using a TAP-tag approach with the purification handle fused to protein Pop4[30] ("Methods"). The isolate contained all expected proteins and RNA components and RNase MRP was active (Supplementary Fig. 1). RNase MRP particles were separated from RNase P during data processing using 3D classification ("Methods").

The resultant map had an overall resolution of 3.0 Å (Supplementary Table 1), with the central regions as good as 2.5 Å (Supplementary Figs. 2 and 3). The final model included all known components of RNase MRP, except for a peripheral protein Pop3. The density corresponding to Pop3 was clearly present, but the map quality in this region was not sufficient for a reliable atomic reconstruction of the Pop3 structure; for

illustrative purposes Pop3 was modeled into the RNase MRP map using its structure in yeast RNase P[21].

Similar to the structures of yeast and human RNases P[21,22], RNase MRP structure (Fig. 2) is dominated by proteins. The proteins are forming a clamp-like structure embedding RNA and protecting most of it from the solvent, consistent with prior biochemical data[30]. The basic patches of RNase MRP proteins largely face the RNA component (Supplementary Fig. 4). The phylogenetically conserved RNA elements corresponding to the catalytic center in RNase P and forming the catalytic center in RNase MRP protrude into the central part of the protein clamp opening and are exposed to the solvent.

Unlike bacterial RNase P RNA[31], RNase MRP RNA is missing elements that can serve to stabilize its three-dimensional organization and, similar to eukaryotic RNase P[21,22], uses protein components as the main structural scaffold (below).

**Structure of RNase MRP RNA.** RNase MRP RNA forms an essentially single-layered structure dominated by coaxially stacked helical regions (Fig. 1c). In the C-domain, helical stem P1 forms a coaxial stack with stem P4, while stems P19, P2, and the proximal part of P3 form a semi-continuous helix connected to

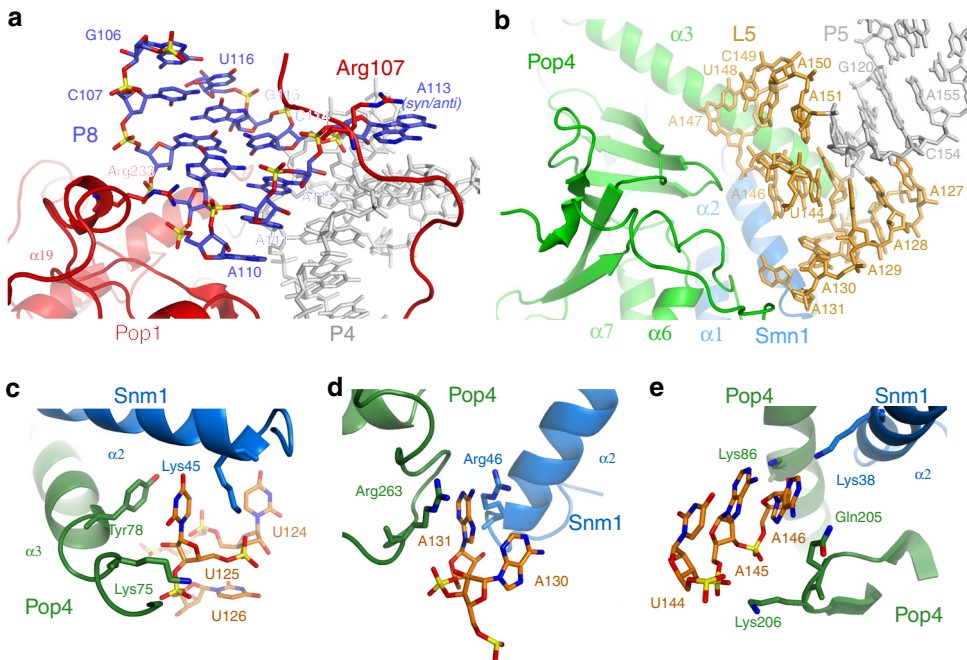

**Fig. 3 RNA-protein interactions in RNase MRP. a** The structure of the conserved element 5′-GA(G/A)A(G/A)-3′ (nucleotides 109–113) and its interactions with protein Pop1 (red). The 5′-GA(G/A)A(G/A)-3′ element folds into a GNRA tetraloop capping stem P8, with the last purine (A113) bulging out. A113 is observed in both syn- and anti- conformations; the syn- conformation appears to be stabilized by a stacking interaction with the conserved Arg107 of protein Pop1. **b–e** Loop L5 of the S-domain caps the P5 stem. The loop is devoid of the base pairs; its structure is stabilized by interactions with protein components Pop4 (green) and Snm1 (blue).

the P1/P4 stack via a four-way junction. The fold of the C-domain of RNase MRP RNA is remarkably similar to the structurally conserved fold of the C-domains found in RNases P from all domains of life[21,22,27,32–34], including yeast RNase P (Fig. 1d).

The key areas forming the catalytic center in yeast RNase P (regions CR-IV, CR-Va, stems P4 and P2), as well as P1 and the proximal part of P3 are folded virtually identically in RNase MRP; the sequence differences found in the central regions of the C-domains of RNases MRP and RNase P do not lead to substantial structural differences. The main structural difference between the C-domains of RNase MRP and RNase P is the orientation of the peripheral P15 stems, which changes by ~45° (Fig. 1c, d) due to interactions with RNase MRP protein Rmp1 (below). Interestingly, P15 stems are absent in human RNase MRP and RNase P[3,4,22].

The A70G mutation in human RNase MRP (corresponding to A84G in *S. cerevisiae*) leads to the developmental disorder Cartilage Hair Hypoplasia (CHH)[14]. A84 forms a reversed Hoogsteen base pair with U314, which is not isosteric to any of the possible GU base pairs; thus, the change in the geometry caused by the transition to GU is the likely cause of the RNase MRP functional deficiency leading to CHH.

The S-domains of RNase MRP and RNase P are structurally different, consistent with the divergent substrate specificities of the two enzymes. The central feature of the RNase MRP S-domain is a coaxial stack of helixes P5 and P8 (Fig. 1c). The distal part of P8 hosts a salient phylogenetically conserved feature of RNase MRP: the 5′-GA(G/A)A(G/A)-3′ element (nucleotides 109–113)[35]. This element folds into a canonical GNRA tetraloop capping the P8 stem, with the fifth nucleotide (A113) bulging out; A113 is observed adopting both anti- and syn- conformations (Fig. 3a and Supplementary Fig. 5). This region is involved in interactions with the protein component Pop1; a stacking interaction with the phylogenetically conserved Arg107 appears to stabilize the syn- conformation of A113 (Fig. 3a). Mutations

that disrupt the equivalent of the P8 stem in human RNase MRP lead to developmental disorders[14].

Helix P5 is closed by a 29 nucleotide-long loop dubbed here L5 (Fig. 1a). The proximal part of the loop folds into a well-ordered complex structure devoid of canonical base pairs and stabilized by extensive interactions with proteins Pop4 and Snm1 (Fig. 3b–e). The distal part of L5 (U132–U143) is disordered in the structure. The sequence of the proximal part of the L5 loop is conserved in Saccharomycetaceae[36], while the sequence of the disordered distal part is not.

The P8/P5 helical stack of RNase MRP is docked against the P1/P4 helical stack of the C-domain in a manner closely resembling the docking of the P8/P9 helical stack against P1/P4 in yeast RNase P (Supplementary Fig. 6): RNase MRP P5 stem occupies a position matching the position of the P9 stem in the S-domain of RNase P, while the position of the RNase MRP P8 stem closely matches the position of the P8 stem in RNase P. Thus, the structural similarity between RNase MRP and RNase P RNAs extends to parts of the S-domains, albeit the similarities in the S-domains are somewhat superficial as the loops closing the matching stems are different in RNase MRP and RNase P, as is the connectivity of the elements (Fig. 1).

Long helical stems P6 and P7 are located in the peripheral part of the S-domain (Fig. 1c). The helical elements of the S-domain are connected by linker regions dubbed here J8/9, J6/7, and J7/9 (Fig. 1a, c). RNase MRP S-domain is connected to the C-domain by a short helical stem dubbed here P9; it should be noted that RNase MRP P9 is not a structural equivalent of the stem P7 in RNase P.

**Proteins in RNase MRP.** Pop1, the largest protein component of RNase MRP, forms the main structural brace in the RNP (Fig. 4a, Supplementary Fig. 7a). The C-terminal part of the protein engages the C-domain RNA, as it does in yeast RNase P[21] (Supplementary Fig. 7b). The structures of the C-domain RNAs in

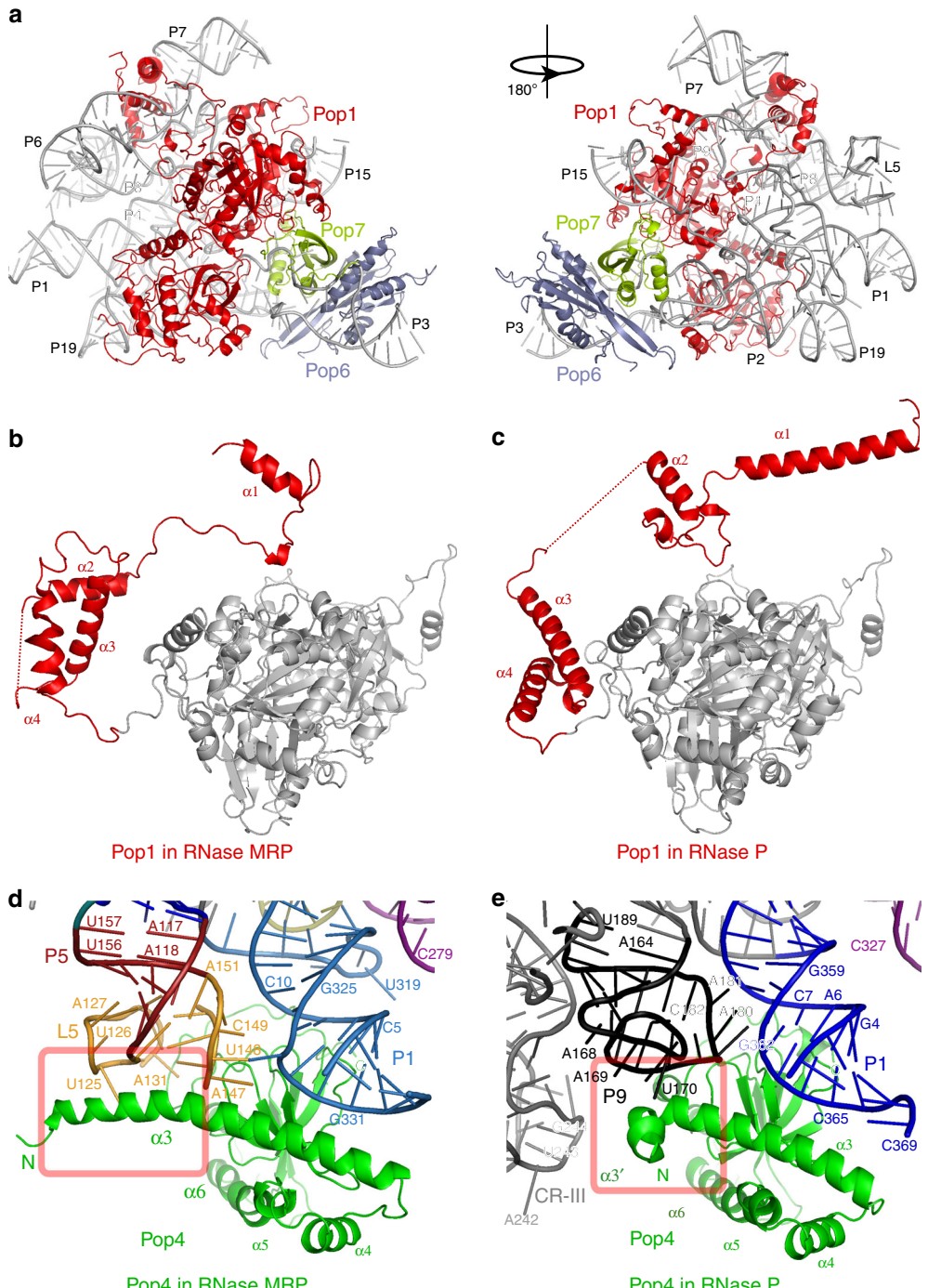

**Fig. 4 Proteins Pop1 and Pop4 stabilize the overall RNase MRP RNA structure and undergo RNA-driven remodeling. a** Interactions of Pop1 (red), Pop6 (gray), and Pop7 (pale green) with RNase MRP RNA (gray). **b**, **c** The fold of the N-terminal part of Pop1 in RNase MRP differs from that in RNase P[21]. The N-terminal parts of Pop1 are shown in red; the similarly folded C-terminal parts of Pop1 are shown in gray. Interactions of Pop4 (green) with the RNA components of RNase MRP (**d**) and RNase P[21] (**e**). RNA elements are marked and color coded according to Fig. 1a, b. The part of the Pop4 α3 helix that folds differently in RNase MRP and RNase P is boxed.

RNase MRP and RNase P are similar (above), and the C-terminal part of Pop1 (residues 203–875) folds and interacts with RNA essentially the same way in the two enzymes (RMSD = 2.6 Å).

At the same time, the N-terminal part of Pop1 engages mostly the S-domain RNA, which is different in RNase MRP and RNase P. The N-terminal part of Pop1 undergoes RNA-driven structural remodeling and adopts distinct conformations in the two RNPs (Fig. 4b, c). RNA-driven structural remodeling of Pop1 can explain

the ability of this protein to stabilize diverse RNPs[17,28], even when the engaged RNA components are structurally distinct. It should be noted that a short stretch of the N-terminal part of Pop1 (residues 91–99) interacts with an area of the C-domain RNA (parts of P4 stem and CR-IV), which adopts the same conformation in RNase MRP and RNase P. In this region the structures of Pop1 in RNase MRP and RNase P converge, only to diverge away from this short stretch (Supplementary Fig. 7c).

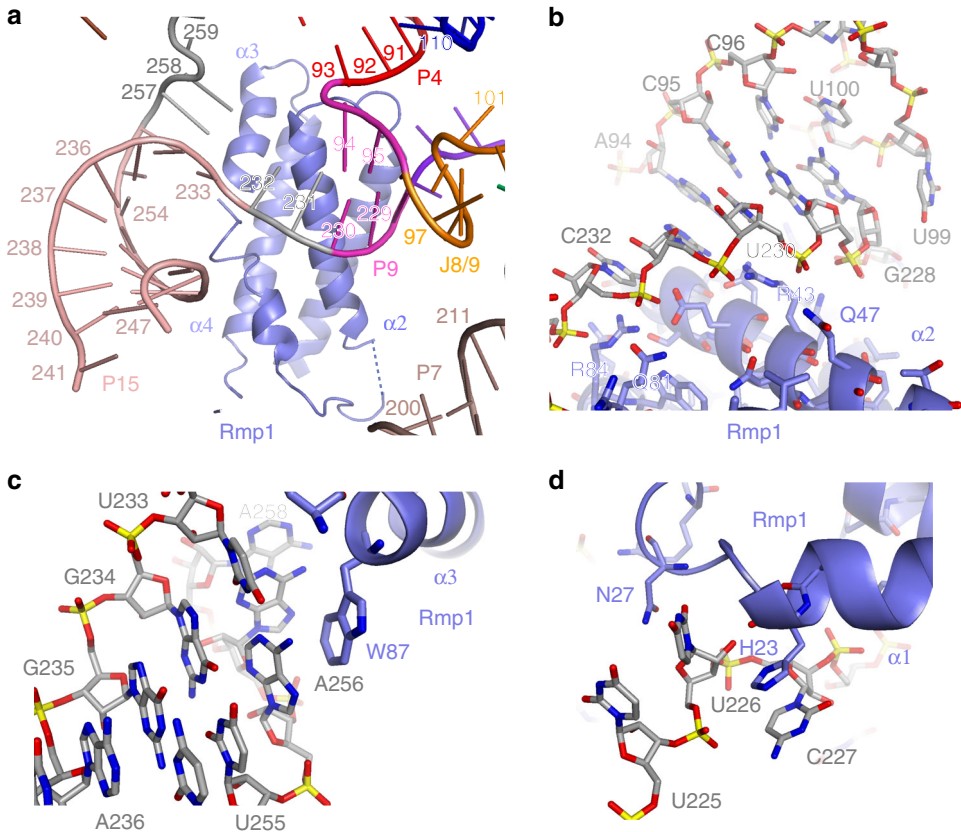

**Fig. 5 Interactions of protein Rmp1 with the RNA component of RNase MRP. a–d** Rmp1 binds at the foundation of the P15 helical stem, in the vicinity of the catalytic center, interacting with stems P4, P9, and the J7/9 junction. Interactions of Rmp1 with RNase MRP RNA result in a change of the P15 stem orientation compared to that observed in RNase P[21].

Proteins Pop6 and Pop7 form a heterodimer that helps to engage Pop1 in both RNase MRP (Fig. 4a) and RNase P[21,22,26,37]. Pop6 and Pop7 interact with P3 RNA regions that adopt similar folds in RNase MRP and RNase P and are folded similarly in the two enzymes (RMSD = 1.8 Å and 1.3 Å, respectively).

Protein Pop4 bridges together stem P1 of the C-domain RNA and the L5 loop of the S-domain (Fig. 4d, Supplementary Fig. 8a). This structural role is similar to the role played by Pop4 in RNase P, where it bridges together stems P1 and P9[21] (Fig. 4e, Supplementary Fig. 8b). The structures of the S-domains of RNase MRP and RNase P differ (above), and the part of Pop4 that engages the S-domain RNA undergoes RNA-driven structural remodeling (Fig. 4d, e, Supplementary Fig. 8c): the long alpha helix α3 found in RNase MRP Pop4 is disrupted in RNase P. In RNase MRP, the N-terminal part of α3 is involved in extensive interactions with the L5 RNA loop, stabilizing its structure (Figs. 3b–e, 4d), whereas in RNase P the corresponding fragment of Pop4 is rotated ~135° and engages stem P9 (Fig. 4e). It should be noted that the extended version of the α3 helix found in RNase MRP would clash with the CR-III element of the RNase P S-domain (Fig. 4d, e). The differences in the S-domains of RNase MRP and RNase P RNAs necessitate the structural remodeling of Pop4; however, the similarity in the positioning of P5/L5 in RNase MRP and stem P9 in RNase P (Fig. 4d, e, Supplementary Fig. 6) allows for the structural divergence to be localized to only the N-terminal region of Pop4, while the rest of the protein adopts essentially the same fold in the two enzymes (RMSD = 2.3 Å for residues 85–279).

Proteins Pop5, Pop8, and two copies of Rpp1 form a heterotetramer that interacts with the C-domain (Supplementary Fig. 9a). The structure of the engaged RNA region is very similar in RNase MRP and RNase P; accordingly, these proteins interact with RNA the same way in the two RNPs and have essentially the same structure (RMSD = 1.7 Å for the heterotetramer), with one notable exception. In RNase P[21], Pop5 interacts with the N-terminal alpha helix α1 of Pop1 (Supplementary Fig. 9c). The fold of the N-terminal part of Pop1 in RNase MRP differs from that in RNase P (above), and Pop1 helix α1 does not reach Pop5, instead engaging protein Rmp1, which is not present in RNase P, and in RNase MRP the structural role of Pop1 helix α1 is played by the helix α6 of Pop5 itself (Supplementary Fig. 9b); α6 appears to be disordered in RNase P[21]. Thus, the RNA-driven structural remodeling of Pop1 leads to the remodeling of another protein component, Pop5.

RNase MRP protein Snm1 has a homolog in yeast RNase P, Rpr2. Both Snm1 and Rpr2 are nestled between Pop4 and Pop3 (Supplementary Fig. 8a, b), interact with Pop4 in a similar fashion, and adopt generally similar folds (Supplementary Fig. 8d, e). Snm1 binds to the S-domain of the RNase MRP RNA and is involved in the stabilization of the structure of the L5 RNA loop (Fig. 3b–e).

RNase MRP protein Rmp1 is a core RNase MRP protein involved in extensive interactions with the RNA component, binding primarily to the foundation of the P15 stem, as well as to the J7/9 region and stem P9 (Fig. 5, Supplementary Fig. 10). Interactions of Rmp1 with the foundation of the P15 stem result in a ~45° change in the stem's orientation compared to that in RNase P. Rmp1 binds in the vicinity of the catalytic center of the enzyme and is involved in the formation of the substrate binding pocket of RNase MRP (below).

The comparison of the protein components of RNase MRP and RNase P demonstrates that RNA-binding protein components

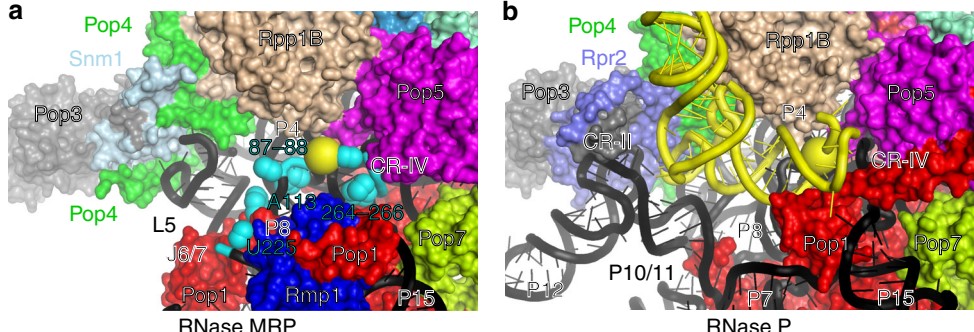

**Fig. 6 Substrate binding pockets in RNase MRP and RNase P.** Protein components are color coded according to Fig. 2. RNase MRP and RNase P RNAs are shown as cartoons in black. The substrate pre-tRNA in the substrate binding pocket of RNase P[21] (**b**) is shown as a cartoon in yellow. The location of the scissile bonds is shown by yellow spheres (**a**, **b**). The locations of the RNase MRP nucleotides that crosslink to substrates[20] are shown as spheres in cyan (**a**).

shared by several RNPs can cope with the divergence in RNA by adopting distinct conformations. This RNA-driven protein remodeling allows for the binding of the same protein to different RNAs that are not necessarily similar at the protein binding sites.

**Substrate binding pocket of RNase MRP.** Despite their close evolutionary relationship and the similarity of the C-domains, RNase MRP and RNase P have orthogonal binding activities, recognizing distinct and nonoverlapping sets of substrates[3,38]. One known exception is the reported in vitro cleavage of a dimeric tRNA precursor pre-tRNA^Ser-Met by *Schizosaccharomyces pombe* RNase MRP[39]; however, in this case RNase MRP apparently recognizes unique features of that specific dimer, as opposed to the RNase P recognition of the characteristic tRNA fold[3]. The divergence of RNase MRP and RNase P specificities is a manifestation of structural differences between the substrate recognizing elements of the two related enzymes.

Photoreactive RNase MRP substrates form UV-induced crosslinks to nucleotides A113, U225 in the S-domain and nucleotides G87-U88, G264-U266 in the C-domain of RNase MRP RNA, as well as to proteins Pop1, Pop4, Pop5, and Rpp1[20]. These elements define the substrate binding pocket of RNase MRP (Fig. 6a); the RNase MRP analog of the RNase P catalytic center (regions CR-IV, CR-Va, stems P4 and P2[21,22,27,32–34]) is also localized to this pocket.

RNase MRP RNA elements lining the substrate binding pocket include S-domain stem P8 with its terminal loop containing the phylogenetically conserved 5′-GA(G/A)A(G/A)-3′ element[35] (which crosslinks to substrates[20]), as well as the S-domain stem/loop P5/L5 and S-domain junction element J6/7 (which also crosslinks to substrates[20]). The elements of the RNA C-domain forming the catalytic center in RNase P (CR-IV, CR-Va, stems P4 and P2[21,22,27,32–34]) are also exposed to the substrate binding pocket of RNase MRP. RNase MRP protein parts contributing to the formation of the substrate binding pocket (Fig. 6a) include the N-terminal segment of Pop1, proteins Pop4, Pop5, Rmp1, and one of the copies of Rpp1. Despite the virtually identical folds of the RNA catalytic centers, the overall topologies of the larger substrate binding pockets in RNase MRP and RNase P are different, in keeping with the divergent specificities of the two related enzymes (Fig. 6).

The CRII/III element and P10/11 stem that contribute to the recognition of the pre-tRNA substrate in RNase P[21,22,27,32–34,40] are missing in RNase MRP. RNase MRP S-domain elements P5/L5, P8, J6/7 define a pocket that is tighter than that in RNase P.

It should be noted that tRNA is not expected to fit into the substrate binding pocket of RNase MRP, mainly due to the steric clash between the "elbow" part of pre-tRNA and the L5 loop of the S-domain (Fig. 6).

In addition to the difference in the S-domains of RNase MRP and RNase P RNAs, the presence of a unique RNase MRP protein Rmp1 near the catalytic center, as well as RNA-driven remodeling of the N-terminal part of protein Pop1 (above) are also major contributors to the divergence of the substrate binding pocket of RNase MRP from that of RNase P and, by inference, to differences in the specificities of the two enzymes. There are no apparent structural indications of any differences in the chemical mechanisms employed by RNases MRP and P.

Our results demonstrate that RNase MRP has acquired its distinct specificity without significant changes in the structural organization of the catalytic center inherited from the progenitor RNase P RNP. Instead, unique auxiliary RNA elements that are introduced into the immediate vicinity of the conserved catalytic center, acting together with RNA-driven remodeling of proteins shared with RNase P, as well as changes in the protein composition, substantially alter the substrate binding pocket of this RNA-based enzyme.

## Methods

**Overview.** Two attempts to determine the structure of RNase MRP have been made. The first attempt relied on the RNase MRP holoenzyme isolated from *S. cerevisiae* using a purification tag fused to RNase MRP protein component Rmp1 as described below. The isolated complex was unstable when stored on ice, which compromised its usefulness for structural studies. Ultimately, the quality of the resulting reconstruction (6.8 Å resolution) was not satisfactory.

The second attempt relied on the samples isolated from *S. cerevisiae* using a purification tag fused to protein Pop4. Since Pop4 is found in both RNase MRP and the related RNase P, the resulting samples contained a mix of the two complexes. RNase MRP and RNase P particles were separated in silico, by 3D classification using the low-resolution RNase MRP reconstruction obtained in the first attempt (above) and the published reconstruction of the RNase P holoenzyme[21] as references. This approach yielded a 3.0 Å reconstruction of the RNase MRP particle.

**RNase MRP isolation.** The RNase MRP holoenzyme was isolated from *S. cerevisiae* strain OE1004[38]. This strain (*MATa RMP1::TAPHIS8::TRP1 sep1::URA3 pep4::LEU2 nuc1::LEU2 ade2-1 trp1-1 his3-11,15 can1-100 ura3-1 leu2-3,112*) has a purification tag (TAPHIS8) fused to the C-terminus of protein Rmp1. The purification tag was similar to the standard TAP tag[41], but with eight histidine residues replacing the calmodulin-binding fragment[38].

The RNase MRP holoenzyme was isolated as previously described[38] with minor modifications. Yeast were grown in 32 l of YPD media at 30 °C to the late logarithmic phase with vigorous aeration. The culture was cooled on ice; the cells (~300 g) were harvested by centrifugation at 4000 × g (4 °C), washed with water twice, followed by a wash with 20% glycerol, frozen in liquid nitrogen, and stored at −75 °C. Frozen cells were thawed in cold water and resuspended in 200 ml of a

buffer containing 20 mM Tris-HCl (pH 7.9), 150 mM KCl, 5 mM Mg-acetate, 10% glycerol, 1 mM PMSF, and 400 mg of deproteinated total yeast RNA (Sigma). The cells were disrupted using a BeadBeater (Biospec) (15 × 15-s pulses on ice), then Tween 20 was added to 0.1% (v/v), and the extract was clarified by centrifugation at 17,000 × $g$ for 1 h (4 °C) followed by ultracentrifugation at 100,000 × $g$ for 3 h (4 °C). The clarified extract was mixed with 3 ml of rabbit IgG agarose (Sigma) and incubated for 5 h at 4 °C with light agitation. The IgG agarose was washed six times with five volumes of the buffer containing 20 mM Tris-HCl (pH 7.9), 150 mM KCl, 1 mM Mg-acetate, 10% glycerol, 1 mM PMSF, and 0.1% (v/v) Tween 20 (Buffer A), and resuspended in 2 ml of the same buffer, supplemented with 10 mg of deproteinated total yeast RNA. Then 80 units of tobacco etch virus (TEV) protease were added and the sample was incubated for 12 h at 4 °C with light agitation. The resin was pelleted by centrifugation at 500 × $g$ for 5 min (4 °C), and the supernatant was collected; the resin was additionally washed twice with 5 ml of Buffer A. The three fractions of the supernatant were combined and cleared of the residual resin by the centrifugation at 4000 × $g$ for 3 min. TEV protease was essentially removed by three rounds of the sample concentration to the final volume of 0.5 ml using an Amicon-Ultra 15 (100 kDa MWCO) concentrator (Millipore), followed by the dilution to the final volume of 4 ml of the Buffer A. The final volume was adjusted to 3 ml, and the sample was incubated with 2 ml of Ni-NTA Agarose (QIAGEN) in buffer A for 5 h at 4 °C with light agitation. The resin was washed six times with 10 ml of Buffer A supplemented with 10 mM Na-imidazole (pH 7.4). After the final wash, the resin was resuspended in a buffer containing 10 ml of 400 mM Na-imidazole (pH 7.4), 100 mM KCl, 2.5 mM Mg-acetate, 5% glycerol, 0.5 mM PMSF, and 0.1% (v/v) Tween 20, and RNase MRP was eluted for 25 min at 4 °C with light agitation. After elution, the residual resin was removed by centrifugation at 4000 × $g$ for 3 min, and the elution buffer was exchanged for a buffer containing 20 mM Tris-HCl (pH 8.0), 75 mM KCl, 75 mM NaCl, 5 mM Mg-acetate, 1 mM TCEP, 0.1 mM EDTA, 0.1 mM PMSF, 0.5% glycerol, and 0.1% (v/v) Tween 20 (Buffer B), and concentrated using an Amicon-Ultra 4 (100 kDa MWCO) concentrator (Millipore). The enzyme concentration was adjusted to ~0.15 pmol/μl as estimated by denaturing gel electrophoresis of the RNA component run alongside with known quantities of reference RNA.

**RNase MRP/RNase P mix isolation.** A mix of RNase P and RNase MRP was isolated from *S. cerevisiae* strain YSW1 (a generous gift from Mark Schmitt, *MATa POP4::TAPTAG::TRP1ks pep4::LEU2 nuc1::LEU2 sep1::URA3 trp1-1 his3-11,15 can-100 ura2-3,112*[10]) using a TAP-tag approach with the tag fused to the C-terminus of protein Pop4. Since Pop4 is a component of both RNase MRP and RNase P, the isolation resulted in the mix of RNase MRP, mature RNase P, and RNase P precursor particles; RNase MRP and RNase P particles were found at ~1:1 molar ratio as judged by denaturing gel electrophoresis of the RNA moieties (Supplementary Fig. 1a). The sample contained all expected proteins (Supplementary Fig. 1b) as verified by mass spectrometry, the RNA components were essentially intact (Supplementary Fig. 1a), and RNase MRP was active as judged by the well-characterized cleavage of a 148 nt-long fragment of the internal transcribed spacer 1 of the rRNA precursor, containing the A3 site[9] (Supplementary Fig. 1c).

The purification procedure described in ref. [30] was used with modifications. Yeast were grown in 32 l of YPD media at 30 °C to the late logarithmic phase with vigorous aeration. The culture was cooled on ice; the cells (~300 g) were harvested by centrifugation at 4000 × $g$ (4 °C), washed with water twice, followed by a wash with 20% glycerol, frozen in liquid nitrogen, and stored at −75 °C. Yeast cells were thawed in cold water and resuspended in a buffer containing 20 mM Tris-HCl (pH 7.9), 150 mM KCl, 10% glycerol, 1 mM PMSF, 1 mM EDTA, supplemented with a protease inhibitor (Pierce). The cells were disrupted using a BeadBeater (Biospec) (15 × 15-s pulses on ice), then Tween 20 was added to 0.1% (v/v), and the extract was clarified by centrifugation at 17,000 × $g$ for 60 min (4 °C) followed by ultracentrifugation at 100,000 × $g$ for 3 h (4 °C). The clarified extract was mixed with 3 ml of rabbit IgG agarose (Sigma) and incubated for 5 h at 4 °C with light agitation. The IgG agarose was washed six times with five volumes of the buffer containing 20 mM Tris-HCl (pH 7.9), 150 mM KCl, 10% glycerol, 1 mM PMSF, 1 mM EDTA, 0.1% (v/v) Tween 20 (Buffer LB), and resuspended in 5 ml of a buffer containing 20 mM Tris-HCl (pH 7.9), 150 mM KCl, 10% glycerol, 1 mM PMSF, 0.1 mM EDTA, 4 mM CaCl₂, 1 mM Na-imidazole (pH 8.0), 5 mM Mg-acetate, and 0.1% (v/v) Tween 20 (Buffer CB). Eighty units of TEV protease and 1600 units of RNasin RNase inhibitor (Promega) were added, and the sample was incubated for 12 h at 4 °C with light agitation. The resin was pelleted by centrifugation at 500 × $g$ for 5 min (4 °C), and the supernatant was collected; the resin was additionally washed twice with 5 ml of Buffer CB. The three fractions of the supernatant were combined; DTT was added to the final concentration of 1 mM. The sample was cleared by centrifugation at 4000 × $g$ for 3 min and 2 ml of calmodulin resin (Agilent Technologies) was added to the sample. The sample was incubated with the resin for 3 h at 4 °C with light agitation. The resin was washed six times with 10 ml of Buffer CB supplemented with 1 mM DTT. After the final wash, the resin was resuspended in 10 ml of a buffer containing 20 mM Tris-HCl (pH 7.9), 150 mM KCl, 10% glycerol, 1 mM PMSF, 10 mM EGTA, 1 mM Na-imidazole (pH 8.0), 5 mM Mg-acetate, 1 mM DTT, and 0.1% (v/v) Tween 20. RNase MRP/RNase P mix was eluted for 2 h at 4 °C with light agitation. After the elution, the elution buffer was exchanged for a buffer containing 20 mM Tris-HCl (pH 7.9), 150 mM KCl,

5 mM Mg-acetate, 1 mM DTT, 0.5% glycerol, 0.1 mM EDTA, 0.1 mM PMSF, and 0.1% (v/v) Tween 20 (Buffer S), and concentrated using an Amicon-Ultra 4 (100 kDa MWCO) concentrator (Millipore). RNase MRP concentration was adjusted to ~1 pmol/μl as estimated by denaturing gel electrophoresis of the isolated RNA run alongside with known quantities of reference RNA. The samples used for vitrification were stored on ice overnight, and vitrified the next day.

**Preparation of continuous carbon grids.** Quantifoil grids (Cu 300 R1.2/1.3 or R2/1, Quantifoil) were coated with a thin continuous layer as follows. In total, 0.12 g of Formvar 15/95 Resin (Electron Microscopy Sciences) was added to 50 ml of 1,2-dichloroethane (Sigma-Aldrich) and allowed to dissolve overnight in a Coplin jar. A 75 mm × 25 mm microscope slide (Ted Pella) was used to gently stir the solution. A new glass slide was dunked in and out of the solution ten times to evenly coat the surface of the slide. The Formvar solution was allowed to dry on the slide a few seconds until no remaining liquid was visible. A razor blade was used to score the edges of the film and the Formvar film was floated onto the surface of water in a small basin. Quantifoil grids were placed top side down on the floating Formvar film. The grids and film were picked up with a piece of printer paper placed on top of them. With the top side of the Formvar-coated grids facing up, the paper was allowed to dry overnight at room temperature on a filter paper in a glass Petri dish. Coins were used to hold down the edges of the paper to prevent it from curling. The Leica EM ACE600 sputter coater was used to apply a thin carbon coating on the Formvar-coated grids. The Leica EM ACE600 reported the final measured thickness of the carbon application of 1.7–2.1 nm, depending on the batch. The paper with the Formvar-coated grids was placed on a filter paper saturated with 1,2-dichloroethane in a covered glass Petri dish for 30 min to dissolve the Formvar. The grids were transferred to another glass Petri dish lined with dry filter paper and allowed to dry before use. The continuous carbon-coated Quantifoil grids were stored in a desiccator and protected from light.

**Generation and processing of the initial cryo-EM data.** RNase MRP was isolated from yeast using a purification handle fused to RNase MRP protein Rmp1 as described above.

The sample was diluted three- to eightfold in the vitrification buffer containing 20 mM Tris-HCl (pH 8.0), 75 mM KCl, 75 mM NaCl, 5 mM Mg-acetate, 1 mM TCEP, 0.1 mM EDTA, and 0.1% (v/v) Tween 20, and centrifuged at 18,000 × $g$ for 5 min at 4 °C. Immediately following the centrifugation, 3.5 μl of the solution was applied to Quantifoil Cu 300 R1.2/1.3 continuous carbon-coated grids prepared as described above and vitrified using Vitrobot Mark IV (FEI) operating at 5 °C and 95% humidity. After the sample loading, the grids were incubated for 15 s, blotted for 3–4 s, and then plunged into liquid ethane.

Cryo-EM data were collected using FEI Titan Krios electron microscope (300 kV). The images were recorded by Falcon 3EC direct electron detector operating in the counting mode at the nominal magnification of ×75,000 (pixel size 0.886 Å, verified with crystal structures of apoferritin). EPU software (FEI) was used for the automated data collection. One image per grid hole was collected at 41 frames per stack. The total exposure rate was 1 e⁻/Å² per frame. The defocus was set to vary in the −1.6 to 3.2 μm range.

The total of 2597 micrographs was collected. Frame alignment and integration were performed as implemented in cisTEM[42]. Excessive particle drift was commonly observed closer to the end of the stacks; accordingly, only the first 33 frames were used. The contrast transfer function (CTF) parameters were estimated using CTFFIND as implemented in cisTEM[42]. Low quality images were discarded, and the remaining 1875 images were used to extract 236,629 candidate particles using an ab initio algorithm implemented in cisTEM. False-positives as well as small and aggregated particles were removed using 2D classification, leaving 71,917 RNase MRP particles. To generate an initial RNase MRP reconstruction, ab initio 3D reconstruction was performed as implemented in cisTEM. The result was used as a starting reference volume in the 3D autorefinement and classification into three 3D classes as implemented in cisTEM. Upon the convergence of the refinement/classification, the single higher resolution class contained 71.3% of the particles. The resolution of the corresponding map (gold standard Fourier shell correlation (FSC) 0.143) was 6.8 Å. The resulting 3D reconstruction of RNase MRP was used as a reference for in silico separation of RNase MRP and RNase P particles, and as a starting reference for the high-resolution refinement as described below.

**Generation and processing of the final cryo-EM data.** The mix of RNase MRP and RNase P holoenzyme particles was isolated from yeast using a purification handle fused to protein Pop4 as described above.

For vitrification, the sample was diluted 12–30-fold in the vitrification buffer containing 20 mM Tris-HCl (pH 7.9), 150 mM KCl, 5 mM Mg-acetate, 1 mM DTT, 0.1 mM EDTA, and 0.1% (v/v) Tween 20, and centrifuged at 18,000 × $g$ for 5 min at 4 °C. Immediately following the centrifugation, 3.5 μl of the solution was applied to Quantifoil Cu 300 R2/1 continuous carbon-coated grids prepared as described above and vitrified using Vitrobot Mark IV (FEI) operating at 5 °C and 95% humidity. After the sample loading, the grids were incubated for 30 s, blotted for 5 s, and then plunged into liquid ethane.

Cryo-EM data were collected using FEI Titan Krios electron microscope (300 kV). The images were recorded by Falcon 3EC direct electron detector operating in the counting mode at the nominal magnification of ×75,000 (pixel size 0.886 Å, verified with crystal structures of apoferritin). EPU software (FEI) was used for the automated data collection. Three nonoverlapping images were collected per grid hole at 39 frames per stack. The total exposure rate was 1 e$^-$/Å$^2$ per frame. The defocus was set to vary in the −1.1 to −2.7 μm range.

The total of 6869 micrographs were collected. All 39 frames in the stacks were aligned and integrated using the Unblur algorithm as implemented in cisTEM[42]. Dose weighting was applied as implemented in cisTEM[42]; the exposure dose for the dose weighting procedure was set to 70% of the actual dose. The CTF parameters were estimated using CTFFIND as implemented in cisTEM[42]; four movie frames were averaged for the CTF estimation. All images and the corresponding CTF estimations were inspected manually and the images judged to be of low quality were excluded from the further data processing. The remaining 4781 images were used to extract 1,653,406 particles using an Ab initio algorithm implemented in cisTEM[42]. False-positive particles were removed using 2D classification, resulting in 1,086,037 particles, which included both RNase MRP and RNase P.

RNase MRP particles were separated from RNase P particles in silico using 3D classification/refinement into two classes as implemented in cisTEM[42]; the resolution limit for the classification was set to 20 Å. The published 3D reconstruction of the RNase P holoenzyme particle[21] was used as the initial reference for one of the 3D classes, and the 6.8 Å 3D reconstruction of the RNase MRP particle (above) was used as the initial reference for the other 3D class. After the convergence of the classification, the RNase MRP class contained 545,483 particles, while the RNase P class contained 540,554 particles, consistent with the ~1:1 RNase MRP RNA to RNase P RNA molar ratio observed for the isolated sample (Supplementary Fig. 1a).

RNase MRP particles were subjected to additional 2D classification and low quality/false-positive classes were removed from the further processing. The resulting 520,656 particles were subjected to 3D autorefinement/classification into two classes as implemented in cisTEM[42]. The initial RNase MRP reconstruction (above), subjected to a 20 Å low-pass filtering was used as the starting reference for the 3D refinement. Upon the convergence of the refinement/classification, the low-resolution class was discarded, and the 3D refinement/classification into two classes was repeated once more. The resultant selection contained 155,205 RNase MRP particles, which were used in the final 3D refinement. It should be noted that a similarly relatively small fraction of the "good" particles was also observed for the related yeast RNase P[21]. The selected RNase MRP particles were used to perform the final 3D refinement as implemented in cisTEM[42] (Supplementary Fig. 2). The 3D reconstruction algorithm implemented in cisTEM allows for the rejection of a specified fraction of the worst fitting particles. The best map was obtained with the rejection of 25% of particles; this map was used for the model building and refinement. The overall resolution of the map was 3.0 Å based on the gold standard FSC of 0.143 criterion (Supplementary Fig. 2d) and was calculated using Phenix 1.18.1 suite[43].

**Model building and refinement.** The published structure of the RNase P holoenzyme[21], the predicted secondary structures of the RNase MRP RNA component[3] and protein components, the crystal structure of proteins Pop6/Pop7 in complex with the P3 subdomain of RNase MRP[37], and the results of the cross-linking analysis of RNase MRP[25] were used as aids in the RNase MRP model building. Overall, the quality of the RNase MRP map (Supplementary Fig. 3) was sufficient for the unambiguous modeling of most parts of the RNA component, and all protein components, except for Pop3, for which density was observed, but the local quality of the map was not sufficient to build a model.

Model building was carried out in Coot 0.8.9[44]. The refinement and validation were performed using Phenix 1.17.1 suite[43]; secondary structure restrains as implemented in Phenix were used in the refinement of the helical regions of the RNA component as appropriate. Protein Pop3 was not included in the final model; for illustrative purposes, the cryo-EM structure of Pop3 as it is found in the context of yeast RNase P[21] was fitted into the corresponding part of the RNase MRP map by maximizing the correlation between the map and the model as implemented in Chimera[45].

**Reporting summary.** Further information on research design is available in the Nature Research Reporting Summary linked to this article.

## Data availability

Cryo-EM map and coordinates were deposited to the Electron Microscopy Data Bank and Protein Data Bank with accession codes EMD-21564 and 6W6V. Other data are available from the corresponding author upon reasonable request.

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

## Acknowledgements

We are grateful to J.-P. Armache, P. Bevilacqua, L. Lindahl, and J. Reese for valuable suggestions, and to M. Schmitt for the generous gift of the YSW1 yeast strain. We are grateful to the staff of the cryo-EM and Proteomics Core facilities at the Huck Institutes of Life Sciences Core for assistance with data collection and processing, and mass spectrometry analysis, respectively. This work was supported by a grant from the National Institutes of Health R01GM135598 to ASK. The cryo-EM facility is supported, in part, by a grant TSF SAP #4100077246 from the Pennsylvania Department of Health.

## Author contributions

A.P. designed experiments, isolated and verified RNPs, vitrified samples, and built the structure; D.L. and H.L. built the structure; C.B. prepared grids, vitrified samples, and collected cryo-EM data; I.B. isolated RNPs; S.L.H. designed experiments and contributed to the implementation of the research; A.S.K. designed experiments, collected and processed cryo-EM data, built the structure, and wrote the paper.

## Competing interests

The authors declare no competing interests.
