## [Peer Review File · Nature Communications]

REVIEWER COMMENTS

Reviewer #1 (Remarks to the Author):

The manuscript by Perederina et al reports the 3Å cryo-EM structure of yeast RNase MRP, a protein-RNA complex that resembles eukaryotic RNase P. This is an important and novel contribution of general interest and is clearly worthy of publication in Nature Communications. RNase MRP is an RNP ribozyme that shares in common the RNA catalytic domain of RNase P, but a unique substrate-specificity domain. Remarkably, most of the proteins associated with RNase P are also associated with MRP, but are arrayed differently and have a unique binding mode with respect to the differing part of the RNA. The manuscript makes extensive references and comparison to the RNase P structure, which greatly adds to its description and our understanding of its significance. The structural determination appears quite sound, and although I was unable to view the coordinates in the PDB, the validation report appears reasonable.

Some minor questions arise:

Are the differences observed in P15 a consequence of different sequences and interactions in the two enzymes, or is P15 simply inherently flexible, and the two structures simply capturing two different conformations available to both enzymes? If the difference is dependent upon the identify of the enzyme, is there any obvious mechanistic consequence?

Based on the differences within the S-domain and similarity of the C-domain, is anything different regarding the chemical mechanism apparent, or is this simply a very interesting example of how evolution has rearranged an enzyme to confer additional activity? Is there any cross-reactivity between MRP and P (i.e., does MRP have any residual pre-tRNA processing capacity, or are the binding activities completely orthogonal)?

The methodology description states that the (previously existing) RNase P map was used to build in the similar parts of the structure. This potentially raises the spectre of model bias. (In this sense I find the Pop3 protein issue more reassuring than

concerning, but a bit more description of how bias was tested for and if present, eliminated, would be reassuring).

I fully appreciate the tour de force this structure represents, but the most important consequence — how substrate discrimination is achieved — remains a bit obscure. Can anything more be said about this, or do we need to await the structure of the enzyme-substrate complex?

Reviewer #2 (Remarks to the Author):

RNase MRP is an important ribozyme involved in rRNA maturation and mRNA metabolism. Here, Pederina and colleagues present the cryo-EM structure of yeast MRP at 3.0 Å resolution, which comes on the heels of archeal, yeast, and human RNase P cryo-EM structures, which are analogous enzymes to RNase MRP but are involved in tRNA maturation. The authors conduct a thorough structural analysis of the RNase MRP structure compared to yeast RNase P, revealing structural similarities surrounding the active site and striking differences surrounding the substrate binding domain. Of note, they observe that while most subunits are shared between RNase MRP and RNase P, they adopt new conformations (e.g. Pop1, Pop4) due to the altered RNA sequence/secondary structure. In addition, they resolve the two RNase MRP specific subunits Smn1 and Rmp1, and can describe their roles in MRP architecture and substrate binding. One major insight comes from the comparison of substrate accessibility, revealing that RNase MRP lacks features required for pre-tRNA binding by RNase P.

Overall, the RNase MRP structure will be of great interest to the RNA ribozyme and metabolism fields, and is thus in principle suitable for publication in Nature Communications. From the figures, it's apparent that the RNase MRP coordinate model can be significantly improved, and this would be a must prior to acceptance. In my view, the paper would benefit from clarifying edits to the figures and shortening of the text or refocusing towards the key points, see below.

Major comments:

1. Fig 4b shows the RNase MRP coordinate model for Pop1, compared to that in panel c of Pop1 in RNase P. The grey part of the protein looks near-identical, yet apparently the MRP model lacks all beta-strands, which are visible in the RNase P model. I would strongly suggest the authors improve their coordinate model and correct secondary structure features. At the reported nominal resolution of 3.0 Å, this should be doable.
2. The authors frequently compare yeast RNase MRP to yeast RNase P. Since the human RNase P structure is also known, a comparison to this would be of interest for the expert reader.
3. The figures and coloring could be improved for clarity throughout. For example, Fig 2 two views would suffice on the top, the bottom views could be replaced with those from the top but with all proteins transparent and the RNA as solid - this would highlight the RNA path, which with the current choice of RNA and protein colors is very difficult to follow. Similarly, in Fig 6, it is very challenging to see the pre-tRNA (maybe add thicker lines?) and labels for example for the active site would be helpful.
4. The text is extremely detailed in the description of structural features, however the importance of these is unclear. While I appreciate that a basic description of the structure is required, the authors could substantially shorten the text without losing their key points on RNA-driven protein remodeling and the RNase MRP substrate binding domain. This will also make the manuscript more accessible to non-experts.

Minor comments:

1. There are multiple mentions of 'electron density' in the text, this is incorrect and should be replaced with cryo-EM density, density, coulomb potential map, or others.
2. The authors should acknowledge the human RNase P structure in the introduction more explicitly.

Reviewer #3 (Remarks to the Author):

Eukaryotic RNase MRP is a ribozyme that acts as an endoribonuclease. It consists of a catalytic RNA core and several protein subunits. The complex is closely related to RNase P, with which it shares most protein subunits but also has unique components and a divergent RNA-subunit. RNase P and RNase MRP process different target RNAs. A cryo-EM structure of RNase P has been published in 2019, but similar structural information is still lacking for the related RNase MRP. This is where the manuscript, by Perederina et al. comes in. They represent the cryo-EM structure of the eukaryotic RNase MRP. The comparison to RNase P reveals how the unique components act as adaptors and how some of the common proteins adapt their conformation to facilitate the contact to the diverging catalytic RNA-subunit.

The manuscript is well written and interesting. It appears largely sound. Minor concerns are:

1) Pixel size "0.886 Å". The quoted pixel size is unusual for a Falcon III at a Krios at 75000 magnification. Many others report pixel sizes for these settings around 1.06 Å. The authors should check, whether they have used the correct parameters.

If the pixel size would be wrong, this could explain the relatively low percentage of "favoured" positions in the Ramachandran plot. If the pixel size would be wrong, the authors should re-refine their model and adapt the resolution claim. However, a wrong pixel size would not have a significant impact on the claims.

If the pixel size is correct, the authors should briefly explain how the pixel size was determined. It is best practice to determine the magnification of the microscope relative to a known X-ray structure. Other standards such as catalase or cross-grating can be easily a few percent off.

2) The distribution of orientations shows a strong preference. Could the authors briefly explain, how this impacted their map or how they mitigated the problem? In extended figure 2 the map looks somewhat distorted, but this is difficult to see.

3) In the methods it is stated that "the exposure dose for the dose weighting procedure was set to 70% of the actual dose". Please add the rational why, this was done.

4) The figures and the legends need some touch-up. The labels ("a", "b", etc.) are generally too large. The colour coding for the subunit changes between figures and even within a composite figure (e.g. Figure 4... Pop 1 is red in a and b and green in c. The same green is also used for Pop6 in a and Pop4 in d (and e)), which make it difficult to follow the representations through the paper.

The figure legends are also very concise and often do not provide sufficient information to understand what is shown in the figures without reading the main text.

We are grateful to the reviewers for their positive and constructive comments and suggestions. Below please find our detailed responses to all of the reviewers' comments.

Changes in the text (in the manuscript file) are highlighted in green.

Reviewers' comments (below) are shown in blue; our responses are shown in black.

Reviewer #1

The manuscript by Perederina et al reports the 3A cryo-EM structure of yeast RNase MRP, a protein-RNA complex that resembles eukaryotic RNase P. This is an important and novel contribution of general interest and is clearly worthy of publication in Nature Communications. RNase MRP is an RNP ribozyme that shares in common the RNA catalytic domain of RNase P, but a unique substrate-specificity domain. Remarkably, most of the proteins associated with RNase P are also associated with MRP, but are arrayed differently and have a unique binding mode with respect to the differing part of the RNA. The manuscript makes extensive references and comparison to the RNase P structure, which greatly adds to its description and our understanding of its significance. The structural determination appears quite sound, and although I was unable to view the coordinates in the PDB, the validation report appears reasonable.

Some minor questions arise:

Are the differences observed in P15 a consequence of different sequences and interactions in the two enzymes, or is P15 simply inherently flexible, and the two structures simply capturing two different conformations available to both enzymes? If the difference is dependent upon the identity of the enzyme, is there any obvious mechanistic consequence?

The following has been added to the revised text:

"The main structural difference between the C-domains of RNase MRP and RNase P is the orientation of the peripheral P15 stems, which changes by $\sim 45^\circ$ (Figs. 1c, d) due to

interactions with RNase MRP protein Rmp1 (below). Interestingly, P15 stems are absent in human RNase MRP and RNase P^[3,4,22].” (page 5)

“Interactions of Rmp1 with the foundation of the P15 stem result in a ~45° change in the stem’s orientation compared to that in RNase P.” (page 9)

Based on the differences within the S-domain and similarity of the C-domain, is anything different regarding the chemical mechanism apparent, or is this simply a very interesting example of how evolution has rearranged an enzyme to confer additional activity?

The revised text contains the following statement:

“There are no apparent structural indications of any differences in the chemical mechanisms employed by RNases MRP and P.

Our results demonstrate that RNase MRP has acquired its distinct specificity without significant changes in the structural organization of the catalytic center inherited from the progenitor RNase P RNP. Instead, unique auxiliary RNA elements that are introduced into the immediate vicinity of the conserved catalytic center, acting together with RNA-driven remodeling of proteins shared with RNase P, as well as changes in the protein composition, substantially alter the substrate binding pocket of this RNA-based enzyme.”
(page 12)

Is there any cross-reactivity between MRP and P (i.e., does MRP have any residual pre-tRNA processing capacity, or are the binding activities completely orthogonal)?

Currently available information indicates that RNase MRP and RNase P substrate specificities are orthogonal. We have added the following to the revised manuscript:

“Despite their close evolutionary relationship and the similarity of the C-domains, RNase MRP and RNase P have orthogonal binding activities, recognizing distinct and non-

overlapping sets of substrates^[3,38]. One known exception is the reported *in vitro* cleavage of a dimeric tRNA precursor pre-tRNA^{Ser-Met} by *S. pombe* RNase MRP^[39]; however, in this case RNase MRP apparently recognizes unique features of that specific dimer, as opposed to the RNase P recognition of the characteristic tRNA fold^[3]. The divergence of RNase MRP and RNase P specificities is a manifestation of structural differences between the substrate recognizing elements of the two related enzymes.” (page 10)

The methodology description states that the (previously existing) RNase P map was used to build in the similar parts of the structure. This potentially raises the spectre of model bias. (In this sense I find the Pop3 protein issue more reassuring than concerning, but a bit more description of how bias was tested for and if present, eliminated, would be reassuring).

The language describing RNase MRP building in the original version was somewhat confusing, and we have revised it as follows:

“The published structure of the RNase P holoenzyme^[20], the predicted secondary structures of the RNase MRP RNA component^[3] and protein components, the crystal structure of proteins Pop6/Pop7 in complex with the P3 subdomain of RNase MRP^[37], and the results of the crosslinking analysis of RNase MRP^[25] were used as aids in the RNase MRP model building.” (page 35)

We did not directly use the RNase P map to build any part of RNase MRP: we used RNase P map only as a temporary aid to superimpose the published RNase P structure and our RNase MRP density. Critically reviewed parts of RNase P model were helpful for building the structure of similarly folded parts of RNase MRP as they allowed to speed the process up, but the RNase MRP model was built directly into the RNase MRP map; the RNase P map was not visible (or needed) during RNase MRP model building. Thus, no RNase P bias was introduced into our RNase MRP model.

I fully appreciate the tour de force this structure represents, but the most important consequence — how substrate discrimination is achieved — remains a bit obscure. Can anything more be said about this, or do we need to await the structure of the enzyme-substrate complex?

We feel more comfortable avoiding speculations about details of substrate recognition until we finish the structure of the enzyme-substrate complex.

Reviewer #2 (Remarks to the Author):

RNase MRP is an important ribozyme involved in rRNA maturation and mRNA metabolism. Here, Perederina and colleagues present the cryo-EM structure of yeast MRP at 3.0 Å resolution, which comes on the heels of archeal, yeast, and human RNase P cryo-EM structures, which are analogous enzymes to RNase MRP but are involved in tRNA maturation. The authors conduct a thorough structural analysis of the RNase MRP structure compared to yeast RNase P, revealing structural similarities surrounding the active site and striking differences surrounding the substrate binding domain. Of note, they observe that while most subunits are shared between RNase MRP and RNase P, they adopt new conformations (e.g. Pop1, Pop4) due to the altered RNA sequence/secondary structure. In addition, they resolve the two RNase MRP specific subunits Smn1 and Rmp1, and can describe their roles in MRP architecture and substrate binding. One major insight comes from the comparison of substrate accessibility, revealing that RNase MRP lacks features required for pre-tRNA binding by RNase P.

Overall, the RNase MRP structure will be of great interest to the RNA ribozyme and metabolism fields, and is thus in principle suitable for publication in Nature Communications. From the figures, it's apparent that the RNase MRP coordinate model can be significantly improved, and this would be a must prior to acceptance. In my view, the paper would benefit from clarifying edits to the figures and shortening of the text or refocusing towards the key points, see below.

The coordinate model has been significantly improved as requested (see below).
Figures and the text have been edited as requested and described below.

Major comments:

1. Fig 4b shows the RNase MRP coordinate model for Pop1, compared to that in panel c of Pop1 in RNase P. The grey part of the protein looks near-identical, yet apparently the MRP model lacks all beta-strands, which are visible in the RNase P model. I would strongly suggest the authors improve their coordinate model and correct secondary structure features. At the reported nominal resolution of 3.0 Å, this should be doable.

We have improved the coordinate model as requested. The new model shows better statistics (please see the updated validation report). Moreover, the secondary structure features (including those of Pop1) are now much more pronounced, addressing the Reviewer's concern regarding Fig. 4b. Figures have been updated.

2. The authors frequently compare yeast RNase MRP to yeast RNase P. Since the human RNase P structure is also known, a comparison to this would be of interest for the expert reader.

We have added additional information on the comparison of yeast RNase MRP with human and archaeal RNases P. The following has been added to the revised text:

“Yeast RNase MRP proteins Pop1, Pop3, Pop4, Pop5, Pop6, Pop7, Pop8 have homologues in human RNase P^[3,4,18,22], while Pop3, Pop4, Pop5, and Rpp1 have homologues in archaeal RNases P^[3,4,27].” (pages 2,3)

“Similar to the structures of yeast and human RNases P^[20, 22], RNase MRP structure (Fig. 2) is dominated by proteins.” (page 4)

“The fold of the C-domain of RNase MRP RNA is remarkably similar to the structurally conserved fold of the C-domains found in RNases P from all domains of life^[20,22,27,32-34], including yeast RNase P (Fig. 1d).” (reference to the human RNase P structure added). (page 5)

“Interestingly, P15 stems are absent in human RNase MRP and RNase P^[3,4,22].” (page 5)

We feel that a more detailed comparison of yeast and human RNase P components, while interesting, is beyond the scope of this work, which is focused on RNase MRP.

We have also included updated information on the components shared by RNase MRP and yeast telomerase. The revised text contains the following:

“RNase MRP proteins Pop1, Pop6, Pop7 are also an essential part of yeast telomerase, where they are involved in the localization of the enzyme and form a structural module that stabilizes the binding of telomerase components Est1 and Est2^[28,29].” (page 3)

3. The figures and coloring could be improved for clarity throughout. For example, Fig 2 two views would suffice on the top, the bottom views could be replaced with those from the top but with all proteins transparent and the RNA as solid - this would highlight the RNA path, which with the current choice of RNA and protein colors is very difficult to follow. Similarly, in Fig 6, it is very challenging to see the pre-tRNA (maybe add thicker lines?) and labels for example for the active site would be helpful.

Figure 2 is modified as suggested. Figures 3, 4 and Supplementary Figures S7, S8, S9 have been modified for color coding consistency; mislabeling in Fig. 4a is corrected. pre-tRNA in Figure 6 is shown by a thicker line, as suggested.

4. The text is extremely detailed in the description of structural features, however the importance of these is unclear. While I appreciate that a basic description of the structure is required, the authors could substantially shorten the text without losing their key points

on RNA-driven protein remodeling and the RNase MRP substrate binding domain. This will also make the manuscript more accessible to non-experts.

As requested, we have shortened the section describing secondary structure features of RNase MRP RNA by removing an exceedingly long paragraph describing features of the C-domain. Please note that the analysis of unique features of RNase MRP S-domain RNA and comparisons between protein folds in the two enzymes are of significant interest to the field, but cannot be easily derived by the majority of interested researchers who do not routinely use specialized structural biology software. Accordingly, we feel that it is important to retain the remaining description in the text; it should be noted that the description left in the revised version of the manuscript is not more detailed than that in the recently published papers describing cryo-EM structures of yeast, human, and archaeal RNases P.

Minor comments:

1. There are multiple mentions of 'electron density' in the text, this is incorrect and should be replaced with cryo-EM density, density, coulomb potential map, or others.

Changed as requested.

2. The authors should acknowledge the human RNase P structure in the introduction more explicitly.

In the revised version we are acknowledging the structure of human RNase P explicitly, as requested. The following has been added to the revised text:

“the structures of *S. cerevisiae* and human RNases P have been determined^[20,22].”

(page 2)

Reviewer #3 (Remarks to the Author):

Eukaryotic RNase MRP is a ribozyme that acts as an endoribonuclease. It consists of a catalytic RNA core and several protein subunits. The complex is closely related to RNase P., with which it shares most protein subunits but also has unique components and a divergent RNA-subunit. RNase P and RNase MRP process different target RNAs. A cryo-EM structure of RNase P has been published in 2019, but similar structural information is still lacking for the related RNase MRP. This is where the manuscript, by Perederina et al. comes in. They represent the cryo-EM structure of the eukaryotic RNase MRP. The comparison to RNase P reveals how the unique components act as adaptors and how some of the common proteins adapt their conformation to facilitate the contact to the diverging catalytic RNA-subunit.

The manuscript is well written and interesting. It appears largely sound. Minor concerns are:

1) Pixel size “0.886 Å”. The quoted pixel size is unusual for a Falcon III at a Krios at 75000 magnification. Many others report pixel sizes for these settings around 1.06 Å. The authors should check, whether they have used the correct parameters.

If the pixel size would be wrong, this could explain the relatively low percentage of “favoured” positions in the Ramachandran plot. If the pixel size would be wrong, the authors should re-refine their model and adapt the resolution claim. However, a wrong pixel size would not have a significant impact on the claims.

If the pixel size is correct, the authors should briefly explain how the pixel size was determined. It is best practice to determine the magnification of the microscope relative to a known X-ray structure. Other standards such as catalase or cross-grating can be easily a few percent off.

The pixel size at a given nominal magnification is unusual because Penn State's Krios is a bespoke instrument configured to serve both Life Science and Material Science communities. Additionally, Penn State's Krios is equipped with a spherical aberration (Cs) corrector, which also affects the pixel size at a given nominal magnification. The pixel size on the object scale has been verified by docking published high resolution crystal structures of apoferritin into an apoferritin cryo-EM map generated at the 75,000x nominal magnification. The pixel size of our instrument at 75,000x magnification was confirmed to be 0.886 Å, as stated originally.

We have improved the coordinate model (see response to Reviewer 1 comments); in the improved model 89% of amino acids are in favoured conformations with no outliers, and the percentage of sidechain outliers is drastically reduced (from 6.1% to 0.3%, please see the "Overall quality at a glance" part of the new validation report).

2) The distribution of orientations shows a strong preference. Could the authors briefly explain, how this impacted their map or how they mitigated the problem? In extended figure 2 the map looks somewhat distorted, but this is difficult to see.

The following was added to the Supplementary Figure 2 legend:

"While there was a substantial orientational bias, it did not have a noticeable effect on the quality of the map. Note that a similar or stronger bias was observed for yeast, human, and archaeal RNases P^[20,22,27]." (page 40)

3) In the methods it is stated that "the exposure dose for the dose weighting procedure was set to 70% of the actual dose". Please add the rational why, this was done.

The commonly used default dose weighting has been optimized using the reconstruction of rotavirus VP6 as a reference (Grant & Grigorieff, eLife, 2015). In our experience, the reconstruction of RNP particles of RNase MRP/P family benefits from a slightly less aggressive dose weighting; in practice, we achieve that by specifying a slightly lower

exposure doze than the actual one. This might be due to a potentially lower sensitivity of these RNPs to radiation damage, or effects of the continuous carbon support that we use, or some other factors, and may or may not be applicable to other similar particles. Given that this minor adjustment does not have any impact on the manuscript's conclusions, and that we have not tested its utility for any other cases, we prefer not to include a detailed rationalization into the text.

4) The figures and the legends need some touch-up. The labels ("a", "b", etc.) are generally too large. The colour coding for the subunit changes between figures and even within a composite figure (e.g. Figure 4... Pop 1 is red in a and b and green in c. The same green is also used for Pop6 in a and Pop4 in d (and e)), which make it difficult to follow the representations through the paper.

The figure legends are also very concise and often do not provide sufficient information to understand what is shown in the figures without reading the main text.

The sizes of the labels ("a", "b", etc.) have been reduced. Figures 3, 4 and Supplementary Figures S7, S8, S9 have been modified for color coding consistency. Figure legends are expanded to make them more informative, as requested.

REVIEWERS' COMMENTS:

Reviewer #2 (Remarks to the Author):

The authors have satisfactorily addressed my concerns, and have improved their figures, model quality, and their structure descriptions. I am happy to support publication.